# The prevalence of *fimA* genotypes of Porphyromonas gingivalis in patients with chronic periodontitis: A meta-analysis

Haini Wang[ORCID]1*, Wenyi Zhang2, Wanchun Wang2, Longmu Zhang3

1 Department of Clinical Laboratory, Qingdao Stomatological Hospital, Qingdao, Shandong, China,
2 Department of Periodontology and Oral Mucosal Diseases, Qingdao Stomatological Hospital, Qingdao, Shandong, China, 3 Department of Clinical Laboratory, Qingdao Blood Center, Qingdao, Shandong, China

* hainiwang@126.com

**Data Availability Statement:** All relevant data are within the paper and its Supporting Information files.

## Abstract

FimA is an important virulence factor of *Porphyromonas gingivalis* (*P. gingivalis*). According to its DNA sequence, the fimA genotype of *P. gingivalis* can be divided into six categories (I, Ib, II, III, IV, V). The fimA gene may be a key factor in the diversity of virulence found in *P. gingivalis*. Moreover, the role fimA plays in the pathogenesis of *P. gingivalis* is closely associated with periodontitis, making it an important factor of study for disease prevention and treatment. In this study, the prevalence of fimA genotypes of *P. gingivalis* in patients with periodontal diseases was evaluated by meta-analysis. The Embase and PubMed databases were searched for articles from 1999 to 2019 using the following search terms: *Porphyromonas gingivalis* or *P. gingivalis;* periodontitis or chronic periodontal disease; fimA or fimA genotype. The reference lists of relevant published articles were searched manually. A total of 17 studies were included in this report. A statistical software package (Stata, version 11.0/mp, StataCorp) was utilized to calculate and analyze the *P. gingivalis* fimA genotypes for each combined incidence estimate. The pooled rates of fimA I, fimA Ib, fimA II, fimA III, fimA IV and fimA V genotypes of *P. gingivalis* were 8.4% (95% CI: 5.7–11.1), 11.7% (95% CI: 7.4–16), 42.9% (95% CI: 34.2–51.7), 6.5% (95% CI: 5.1–7.9), 17.8% (95% CI: 9.0–26.5), and 3.2% (95% CI: 1.6–4.9), respectively. This study showed that the fimA II and fimA IV genotypes of *P. gingivalis* are highly present in patients with periodontal disease. Therefore, these two genotypes may be related to the pathogenesis and progress of periodontal disease, one of the main risk factors of periodontitis.

## Introduction

Periodontitis is the chronic inflammation of periodontal tissue caused by an imbalance between local bacterial infection and host immune response, which leads to the formation of periodontal pockets. Periodontitis is characterized by the progressive loss of gum and alveolar bone absorption. Without treatment, this chronic inflammation can lead to the loss of affected teeth [1].

**Funding:** The authors acknowledge the funding support of Qingdao Key Health Discipline Development Fund 2020-2022 to HW. The funder had no role in study design, data collection and analysis, decision to publish, or preparation of the manuscript.

**Competing interests:** The authors have declared that no competing interests exist.

The major microorganisms associated with chronic periodontal disease are *Porphyromonas gingivalis*, *Aggregatibacter actinomycetemcomitans*, *Treponemadenticola*, *Prevotella nigrescens*, and *Fusobacterium nucleatum* [2]. The most common bacterium related to periodontitis is *Porphyromonas gingivalis* (*P. gingivalis*), a gram-negative anaerobe. *P. gingivalis* colonizes in the mouth and is closely linked to chronic periodontitis [3–5].

Many virulent factors contribute to *P. gingivalis*' ability to affect the gums; proteases (collagenases, gingipains, hemolysin, trypsin), lipopolysaccharides, its capsule, fimbriae, and hemagglutinins all work in parallel to colonize the host tissue [6–8]. However, studies point to fimbriae as the leading virulence factor of *P. gingivalis* [9, 10]. According to the nucleotide sequence differences of the fimA gene, which encodes the fimA protein subunit, the fimA gene is divided into six genotypes: I, Ib, II, III, IV, and V [11]. Many animal experiments and clinical studies have shown that *P. gingivalis* with different fimA genotypes resulted in different virulent characteristics; genotypes I, III, and V are non-toxic, while fimA II, Ib and IV are highly toxic [11–13].

Due to the high prevalence of periodontitis and the myriad of complications that patients face, determining the risk factors that influence the incidences of this disease is crucial to prevention and management strategies. FimA genotypes of *P. gingivalis* have been extensively researched and reports have indicated that fimA genotypes are closely connected with periodontitis. Therefore, it is of important to investigate the prevalence of fimA genotypes of *P. gingivalis* as they may provide a novel approach in the prevention and treatment of periodontitis. Accordingly, a meta-analysis is important for identifying the prevalence of fimA genotypes in periodontal patients [3, 5, 7]. Using a meta-analysis vs a systematic review has the advantages of a larger sample size and higher resolution by combining different study statistics, and is also a comprehensive, accurate, and effective way of understanding the subject under investigation [14]. In our study, articles obtained from Embase and PubMed databases from 1999 to 2019 were analyzed using a statistical software package. The aim of this study is to provide a comprehensive summary of the measures to explore the prevalence of fimA genotypes of *P. gingivalis* in patients with periodontal disease.

## Materials and methods

### Literature search

The Embase and PubMed databases were searched from 1999 to January 2019 using the following search terms: *Porphyromonas gingivalis* or *P. gingivalis*; periodontitis or chronic periodontal disease; fimA or fimA genotype. The reference lists of relevant published articles were searched manually.

### Study selection

167 articles with the keywords were included in the initial list and 150 unrelated studies were excluded. In addition, all studies reporting a fimA genotype of *P. gingivalis* in chronic periodontal disease were reviewed. Studies were included only if they met the following criteria: (a) studies were written in English or Chinese; (b) studies were from between 1999 and 2019; (c) studies reported the rate of fimA genotypes of *P. gingivalis* among chronic periodontal disease. Studies were excluded using the following criteria: (a) studies were reviews, case reports, or comments; (b) studies were non-human studies; (c) studies were duplicates.

### Data extraction and methodological assessment

Relevant information was extracted, including the year of publication, first author's name, sample size, total number of cases, and number of positive numbers. Adjusted point estimation

**Table 1. Summary of data from selected articles on the fimA genotypes of *P. gingivalis* found in periodontitis.**

| First author, published year | Language | Quality assessment score | Sample size | FimA genotype | | | | | |
|---|---|---|---|---|---|---|---|---|---|
| | | | | I | Ib | II | III | IV | V |
| Amano A,1999 [7] | English | 6 | 73 | 4 | Nd | 43 | 5 | 9 | 0 |
| Amano A,2000 [15] | English | 6 | 121 | 8 | Nd | 80 | 7 | 35 | 21 |
| Beikler T,2003 [16] | English | 7 | 102 | 26 | 0 | 39 | 5 | 19 | 4 |
| Guo YH,2005 [17] | Chinese | 7 | 89 | 3 | 0 | 22 | 6 | 25 | 16 |
| Zhao L,2007 [12] | English | 7 | 94 | 8 | 9 | 23 | 6 | 20 | 3 |
| Yang BT,2016 [18] | Chinese | 8 | 24 | 0 | 3 | 10 | 3 | 4 | 3 |
| Enersen M,2008 [19] | English | 7 | 82 | 3 | 4 | 28 | 8 | 17 | 1 |
| Missailidis CG,2004 [20] | English | 7 | 135 | 6 | 33 | 53 | 9 | 7 | 0 |
| Teixeira SR, 2009 [21] | English | 5 | 152 | Nd | Nd | 47 | Nd | 117 | Nd |
| Davila-Perez C, 2007 [22] | English | 8 | 25 | 3 | 5 | 4 | 1 | 1 | 0 |
| Pérez-Chaparro PJ,2009 [5] | English | 7 | 30 | 2 | 5 | 16 | 3 | 4 | 0 |
| Sandra Moreno, 2015 [23] | English | 7 | 26 | 3 | 2 | 15 | 2 | 0 | 0 |
| Feng X,2013 [24] | English | 7 | 55 | 9 | 11 | 22 | 3 | 3 | 0 |
| Ji-Hoi Moon,2013 [3] | English | 7 | 277 | 36 | 49 | 191 | 19 | 43 | 14 |
| Miriam Puig-Silla,2012 [25] | English | 7 | 33 | 4 | 4 | 13 | 3 | 5 | 0 |
| Asano H,2003 [26] | English | 5 | 32 | 3 | Nd | 15 | 2 | 5 | 2 |
| Feng X,2014 [27] | English | 7 | 39 | 4 | 9 | 20 | 2 | 1 | 0 |

*NT = not tested.

with 95% confidence intervals was required for quality assessment (Table 1). Data extraction was completed by two authors. Disagreements were resolved by a third reviewer.

The authors assessed data quality using an appropriate quality assessment tool [28]. We assessed each article's quality using eight criteria: (1) definition of the target population; (2) representative probability sampling; (3) sample characteristics matching the population; (4) adequate response rates; (5) standardized data collection methods; (6) reliable survey measures; (7) effective survey measures; (8) appropriate statistical methods. The criteria score was either 0 or 1, which stood for "non conformity" or "conformity", respectively. The total score for each study was between 0 and 8.

## Statistical analysis

We used a statistical software package (Stata, version 11.0/mp, StataCorp) to analyze fimA genotypes of *P. gingivalis* for each combined incidence estimate. The comprehensive prevalence estimate and 95% confidence interval (CI) were determined based on the random or fixed effect model. Considering the possibility of heterogeneity between the studies, the Q-test ($P<0.10$ was considered as significant heterogeneity) and $I^2$ test (values 25, 50, 75%) were used to represent low, medium, and high heterogeneity, respectively. Due to high heterogeneity ($P<0.10$), the random effect model was used for statistical analysis. Furthermore, publication bias was assessed by Egger's and Begger's tests ($P<0.05$ represents significant publication bias).

## Results

### Study selection and characteristics of eligible studies

A total of 167 studies were originally selected through the search process. These articles were identified based on abstracts, titles, and full texts, and after the exclusion process a total of 17

papers were selected for our study. The total sample size was 1389 individuals, including samples from different countries. The search process is demonstrated in Fig 1.

## Results of the meta-analysis

In the 17 included studies, the sample size was 1389 periodontal patients from articles published between 1999 and 2016. Studies were divided into two cohorts (1999–2010 and 2011–2016) for group analysis. To begin there were four main early genotypes: fimA I, fimA II, fimA III, fimA IV and fimA V. Over time, the fimA Ib subtype appeared (Table 1).

## Correlation of fimA genotypes and periodontitis

It was noticed that *P. gingivalis* with various fimA genotypes were present from chronic periodontitis patients. The pooled rates of fimAI, fimA Ib, fimA II, fimA III, fimA IV, and fimA V *P. gingivalis* were 8.4% (95% CI:5.7–11.1; Fig 2), 11.7% (95% CI: 7.4–16; Fig 3), 42.9% (95% CI: 34.2–51.7; Fig 4), 6.5% (95% CI: 5.1–7.9; Fig 5), 17.8%(95% CI: 9.0–26.5; Fig 6), and 3.2% (95% CI: 1.6–4.9; Fig 7) respectively. As we can see in Figs 3 and 6, the prevalence of fimA II and fimA IV *P. gingivalis* were statistically significant compared to others. In general, the fimA II and fimA IV genotypes contributed to a elevated risk of periodontitis.

## Publication bias

When the Begg's funnel plots of the prevalence of the fimA I, Ib, III, and V genotypes of *P. gingivalis* in chronic periodontal patients were examined, no sign of publication bias was observed. In fact, most studies were located in the funnel plots, and therefore the results of all related studies were included in the analysis ($P = 0.005$) (Fig 8).

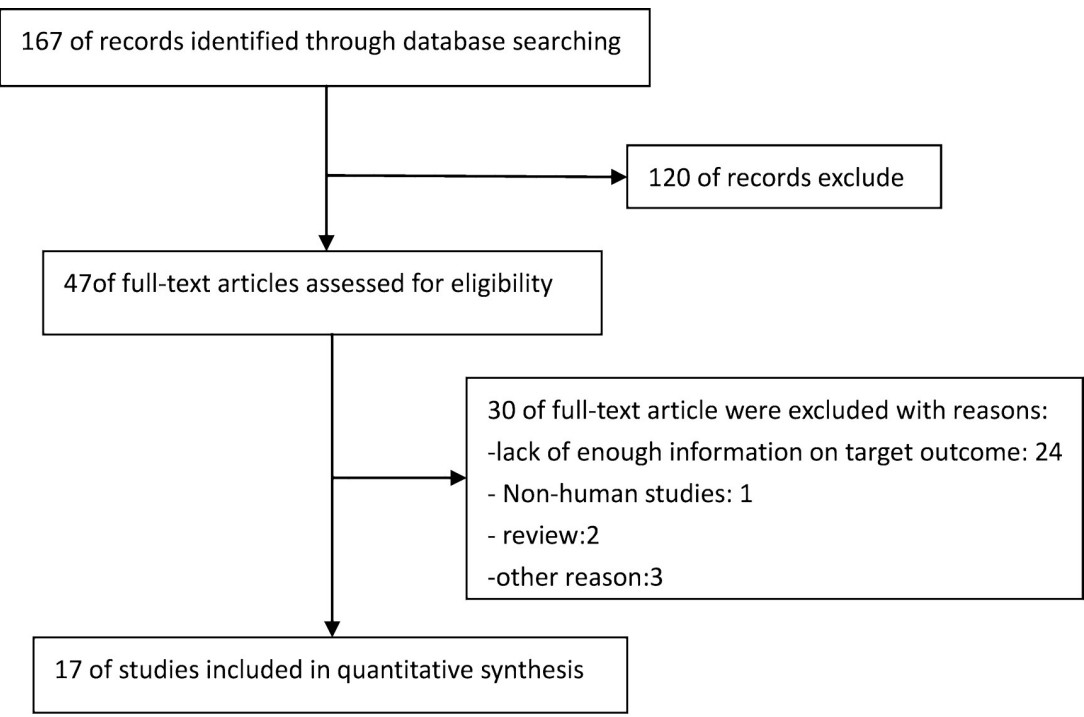

**Fig 1. Flow diagram of studies search.**

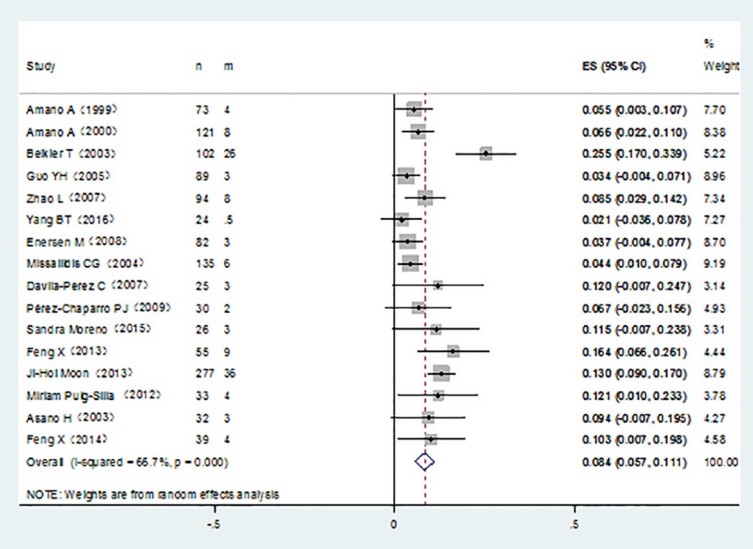

**Fig 2. Forest plot of the prevalence of the fimA I genotype of *P. gingivalis* in periodontitis.**

When examining the Begg's funnel plot of the prevalence of the fimA II genotype of *P. gingivalis* in chronic periodontal patients, signs of publication bias were observed ($P = 0.005$) (Fig 8).

## Discussion

Previous studies indicated that fimA genotypes of *P. gingivalis* are closely connected with periodontitis. In this meta-analysis, we examined the presence of *P. gingivalis* with various fimA genotypes in chronic periodontitis patients. We found that the fimA II and fimA IV genotypes of *P. gingivalis* were the two most predominant genotypes [7, 12, 15–19, 21, 24–27]. There are several plausible reasons that may explain why the fimA II and fimA IV genotypes were more

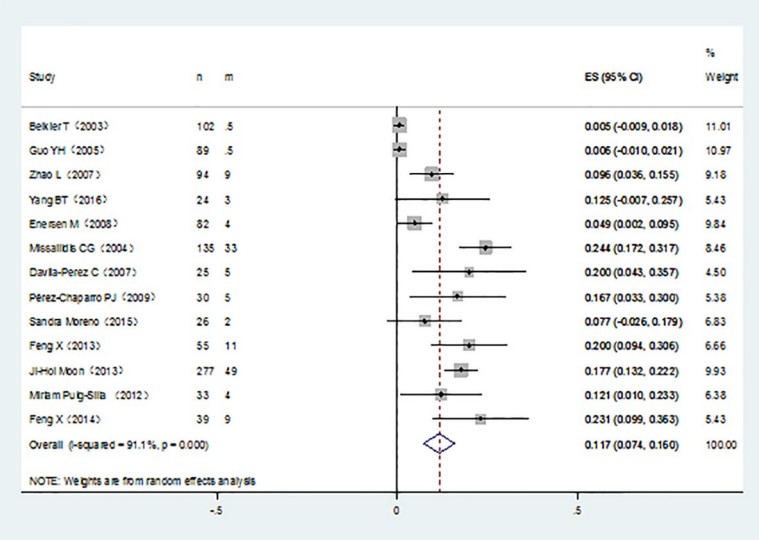

**Fig 3. Forest plot of the prevalence of the fimA Ib genotype of *P. gingivalis* in periodontitis.**

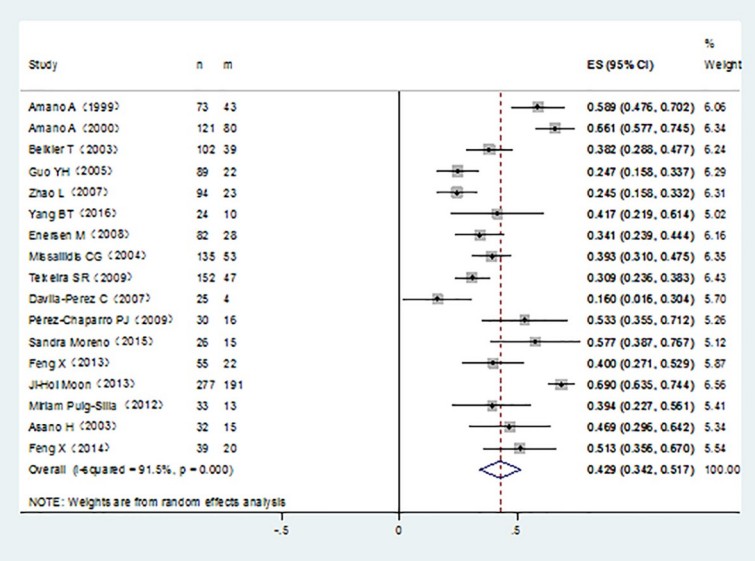

**Fig 4. Forest plot of the prevalence of the fimA II genotype of *P. gingivalis* in periodontitis.**

predominant in this study. Periodontitis is a chronic inflammatory disease that leads to persistent, low-level, systemic inflammation with elevated levels of circulating inflammatory markers [29]. Various studies have indicated that the *P. gingivalis* fimA protein could initiate an immune inflammatory response through a variety of receptors, signal pathways, and cytokines, which enable periodontal tissue cells to express proteolytic enzymes, such as matrix metallo-proteinase (MMP). These proteolytic enzymes participate in the destruction of periodontal connective tissue and alveolar bone absorption, leading to bone defects [30–35]. The fimA II genotype had a greater ability to adhere and invade epithelial cells in mouse models [36] and in vitro studies showed that fimA II could induce higher levels of IL-6 and IL-β than fimA I

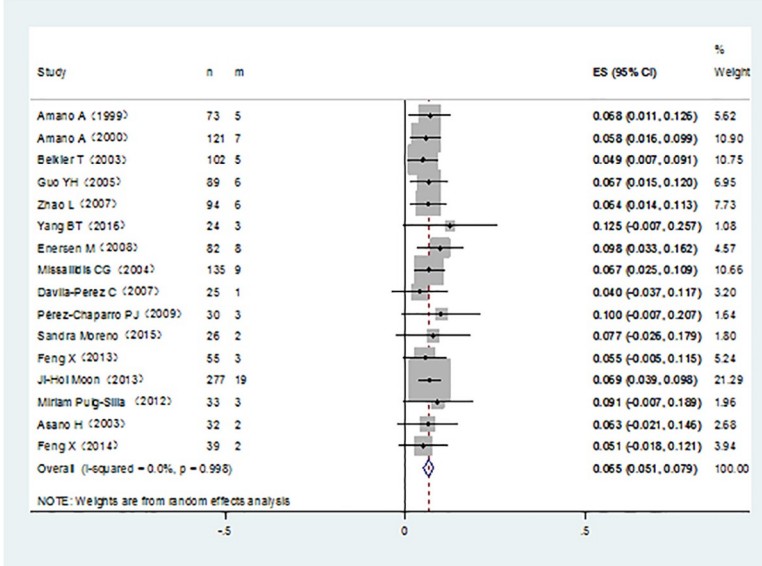

**Fig 5. Forest plot of the prevalence of the fimA III genotype of *P. gingivalis* in periodontitis.**

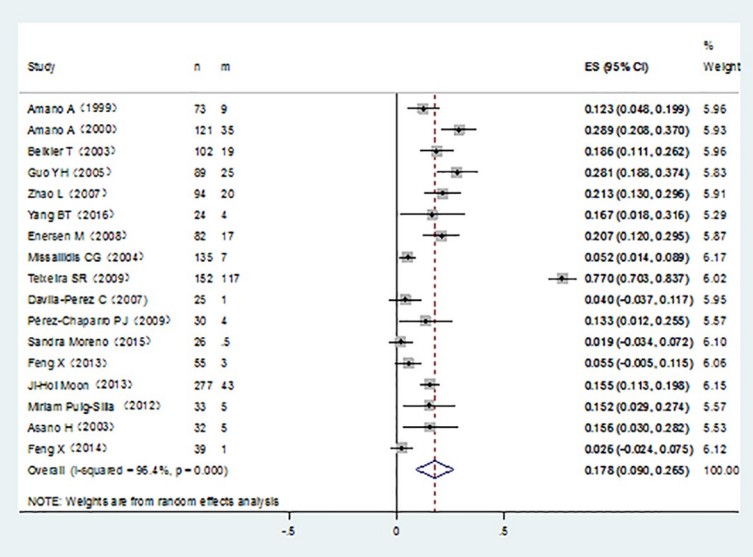

**Fig 6. Forest plot of the prevalence of the fimA IV genotype of *P. gingivalis* in periodontitis.**

[37]. Compared with fimA I, the fimA IV genotype was highly invasive, and this invasiveness was increased in gingival fibroblasts if there was the substituting of allele fimA IV for allele fimA I [38].In addition, fimA II and fimA IV genotypes of *P. gingivalis* could induce the expression and secretion of MMP-8 and MMP-9 in neutrophils in vivo and in vitro [39, 40]. On the contrary, some studies have implicated that the incidence of fimA I and fimA Ib genotypes of *P. gingivalis* are also high [3, 16, 22–23]. A previous study indicated that the fimA I genotype was significantly associated with healthy periodontal people [15]. Further investigation found that fimA Ib had a high sequence homology with fimA I, but their distribution profiles were different [41]. Statistical analysis in the same study showed a significant relationship

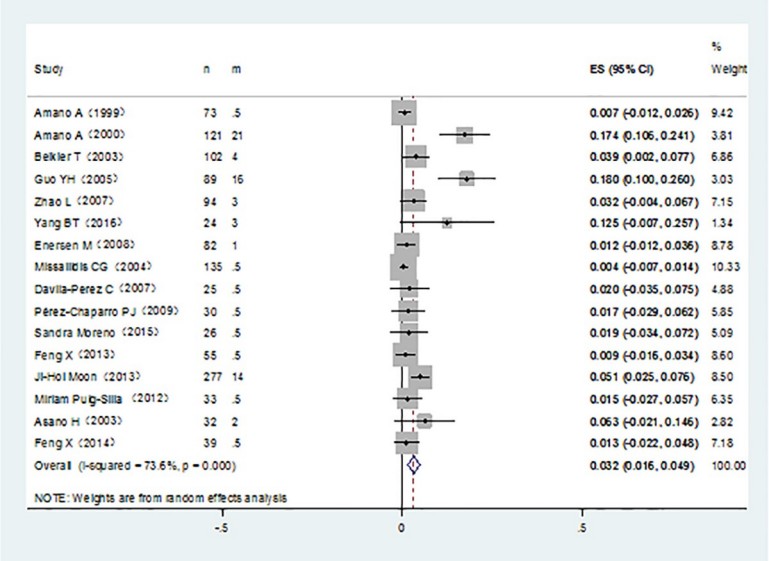

**Fig 7. Forest plot of the prevalence of the fimA V genotype of *P. gingivalis* in periodontitis.**

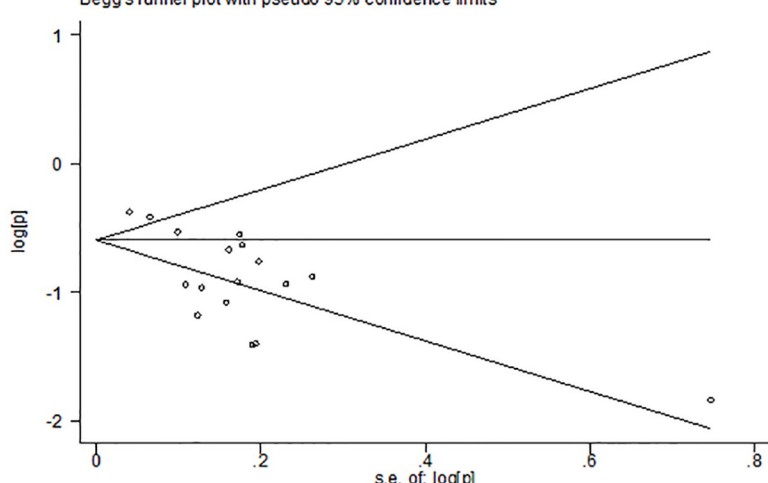

**Fig 8. Begg's funnel plot of the prevalence of the fimA II genotype of *P. gingivalis* in periodontitis.**

between the fimA Ib genotype and periodontitis [41]. The variations of fimA play a major definitive factor in the virulence and pathogenicity of *P. gingivalis*, in relation to the specific proteases, capsulation, and other factors that influence pathogenicity [42–44]. Presently, there is no other meta-analysis similar to this research in terms of ability to compare results. To the best of our knowledge, we identified the fimA genotypes of *P. gingivalis*, which is one of the main etiological factors of chronic periodontal disease. Exploration of the part these genotypes play in this disease can lead to the development of new treatment and prevention methods in the future. In recent years, the research on reducing the pathogenicity of *P. gingivalis* and preventing periodontal disease through the inhibition of fimA protein function has attracted much attention. Researchers have cloned an IgG monoclonal antibody from purified fimA protein and produced the monoclonal antibody using rice suspension [45]. These results suggest that the specific antibodies produced can be used for passive immunization to prevent *P. gingivalis* induced periodontal disease, but further research is needed. Recent studies have shown that photo-activated disinfection mediated by photosensitizer toluidine blue has a significant inhibitory effect on the formation of *P. gingivalis* biofilm. Because of this, the fimA gene is a suitable target for interactions with photo-activated disinfection [46, 47]. There were several limitations in this study that must be addressed. First, publication bias is an inevitable problem in a meta-analysis process, negative trials are sometimes less likely. Considering that our data were extracted from studies written in English and Chinese, which should reduce publication bias to some extent. Second, we were unable to avoid some influential factors, such as smoking, alcohol, age, and gender on account of inadequate data These factors may influence the prevalence of fimA genotypes of *P. gingivalis* and incidence of chronic periodontitis by affecting the ability of *P. gingivalis* to invade the gingival tissue and influence the malignant process. Third, the demographic features of the populations in these studies could have an impact on the results. Finally, some studies that examined the prevalence of fimA genotypes of *P. gingivalis* in periodontitis were not available.

In conclusion, the fimA gene is *P. gingivalis'* main virulence factor, it is important to identify the different fimA genotypes and their prevalence in chronic periodontitis. This study showed that the fimA II and fimA IV genotypes were present at increased levels in patients with periodontal disease. Therefore, these two genotypes may be related to the pathogenesis and progression of periodontal disease and can be considered as the main potential risk factors

of periodontal disease. These results indicate that the pathogenicity of fimA genotypes of *P. gingivalis* needs to be further studied.

## Supporting information

**S1 Checklist. PRISMA checklist.**
(DOC)

## Acknowledgments

The authors wish to thank Professor Dongmeng Qian and Dr. Qin Zheng for their help in revising the manuscript.

## Author Contributions

**Formal analysis:** Longmu Zhang.

**Resources:** Haini Wang.

**Software:** Wenyi Zhang, Longmu Zhang.

**Supervision:** Wenyi Zhang, Wanchun Wang.

**Writing – original draft:** Haini Wang.

**Writing – review & editing:** Haini Wang.

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
