## [Decision Letter · Decision Letter 0]

28 May 2020

PONE-D-20-06291

The prevalence of fimA genotype of Porphyromonas gingivalis in patients with chronic periodontitis :a meta-analysis

PLOS ONE

Dear Dr. Wang,

Thank you for submitting your manuscript to PLoS ONE. After careful consideration, we feel that it has merit but does not fully meet PLOS ONE’s publication criteria as it currently stands. 

After having intensively reviewed your draft, our external referees have indicated major drawbacks. Moreover, our reviewers strongly differed with their final recommendations, and, thus, I have invited a further external referee, to come to a more balanced decision. All in all, the indicated shortcomings are considered reasonable with regard to both PLoS ONE's quality standards and our readership's expectations. Therefore, we invite you to submit a revised version of the manuscript that addresses the points raised during the review process.

We look forward to receiving your revised manuscript.

Kind regards,

Andrej M Kielbassa, Prof. Dr. med. dent. Dr. h. c.

Academic Editor

PLOS ONE

3. Please remove your figures from within your manuscript file, leaving only the individual TIFF/EPS image files, uploaded separately.  These will be automatically included in the reviewers’ PDF.

4. Please include your tables as part of your main manuscript and remove the individual files. Please note that supplementary tables (should remain/ be uploaded) as separate "supporting information" files

Reviewers' comments:

Reviewer's Responses to Questions

**Comments to the Author**

1. Is the manuscript technically sound, and do the data support the conclusions?

Reviewer #1: No

Reviewer #2: Yes

Reviewer #3: Yes

2. Has the statistical analysis been performed appropriately and rigorously? 

Reviewer #1: Yes

Reviewer #2: Yes

Reviewer #3: Yes

3. Have the authors made all data underlying the findings in their manuscript fully available?

Reviewer #1: Yes

Reviewer #2: Yes

Reviewer #3: Yes

4. Is the manuscript presented in an intelligible fashion and written in standard English?

Reviewer #1: No

Reviewer #2: Yes

Reviewer #3: Yes

5. Review Comments to the Author

Reviewer #1: General remark

- English remains a concern, and authors are strongly encouraged to thoroughly revise their draft. Please check grammar and spelling. Please seek some help from a native speaker, to revise, and to facilitate reading.

Abstract

- "Therefore, the two genotypes of Porphyromonas gingivalis appear to contribute to the pathogenesis and progress of periodontal disease, one of the main risk factors of periodontitis." How can you say this? Please provide a sound conclusion, only based on your outcome.

Intro

- "Plaque is the initiating factor of periodontal disease." Please note that "plaque" is a dental term. Use "biofilm" instead, and add a definition.

- Please revise writing, see: "aggregatibacte,Actinomycetemcomitans,Treponemadenticola, Prevotella nigrescens,and Pusobacterium nucleatum". Incorrect spelling is not acceptable with a scientific paper.

- Please elaborate your aims, and give your objectives.

- Deduce a sound and indisputable null hypothesis from your foregoing elaboration.

Meths

- Please expand this section by given more details.

Results

- Again, please expand this section, to facilitate reading.

- Please note that it is your task to guide the readers. Do not leave the latter alone with your graphs. Provide a neutral explanation.

Disc

- The same with this section. Please note that your Discussion has not been convincingly elaborated.

Concl

- "Therefore, the two genotypes of Porphyromonas gingivalis are related to the pathogenesis and progress of periodontal disease." Please see comments given above.

Refs

- Following the Authors' Guidelines is strongly recommended.

In total, although covering an interesting topic, this draft would not seem worth following without a stringent and thorough revision.

Reviewer #2: Review Comments to the Author

"Nice paper"

Reviewer #3: Please check word spacing, organism names and syntax errors in the manuscript. No other major corrections is noted. All the relevant data has been mentioned and the manuscript written in intelligible fashion.

6. PLOS authors have the option to publish the peer review history of their article (what does this mean?). If published, this will include your full peer review and any attached files.

Reviewer #1: No

Reviewer #2: No

Reviewer #3: No

---

## [Author Response · Author response to Decision Letter 0]

12 Jun 2020

Please let me know if you have any information about the manuscript. thank you.

---

## [Decision Letter · Decision Letter 1]

29 Jun 2020

PONE-D-20-06291R1

The prevalence of fimA genotype of Porphyromonas gingivalis in patients with chronic periodontitis :a meta-analysis

PLOS ONE

Dear Dr. Wang,

Thank you for submitting your manuscript to PLOS ONE. After careful consideration, we feel that it has merit but again does not meet PLOS ONE’s publication criteria as it currently stands.

Having intensively reviewed your draft, our external referees still have indicated major shortcomings. Moreover, our reviewers strongly differed with their final recommendations, and, thus, I have invited a further external referee, to come to a more balanced decision. All in all, the indicated shortcomings are considered reasonable with regard to both PLOS ONE's quality standards and our readership's expectations.

Therefore, we invite you to submit a revised version of the manuscript that addresses each and every point raised during the review process. Please note that a further non-convincing revision (not considered acceptable with regard to language, content, and/or Authors' Guidelines) will lead to outright reject.

We look forward to receiving your revised manuscript.

Kind regards,

Andrej M Kielbassa

Academic Editor

PLOS ONE

Reviewers' comments:

Reviewer's Responses to Questions

**Comments to the Author**

1. If the authors have adequately addressed your comments raised in a previous round of review and you feel that this manuscript is now acceptable for publication, you may indicate that here to bypass the “Comments to the Author” section, enter your conflict of interest statement in the “Confidential to Editor” section, and submit your "Accept" recommendation.

Reviewer #1: (No Response)

Reviewer #2: All comments have been addressed

Reviewer #3: All comments have been addressed

2. Is the manuscript technically sound, and do the data support the conclusions?

Reviewer #1: No

Reviewer #2: Yes

Reviewer #3: Yes

3. Has the statistical analysis been performed appropriately and rigorously? 

Reviewer #1: Yes

Reviewer #2: No

Reviewer #3: Yes

4. Have the authors made all data underlying the findings in their manuscript fully available?

Reviewer #1: Yes

Reviewer #2: Yes

Reviewer #3: Yes

5. Is the manuscript presented in an intelligible fashion and written in standard English?

Reviewer #1: No

Reviewer #2: Yes

Reviewer #3: Yes

6. Review Comments to the Author

Reviewer #1: General remarks

- This revised draft has been improved to some extent, no doubt. However, there are still minor and major drawbacks, and this re-submitted version is not considered ready to proceed.

- Again, the authors are strongly encouraged to seek help from a native speaker experienced with scientific writing.

Title

- "The prevalence of fimA genotype of Porphyromonas gingivalis in patients with chronic periodontitis :a meta-analysis". Please revise for correct punctuation. Remember that it is not considered the reviewers' task to co-author and/or re-edit your submitted draft.

- Go for italics with your species names.

- The term "prevalence of fimA genotype of Porphyromonas gingivitis" would seem misleading. Remember that you did not study other genotypes than "fimA". (Indeed, you have focussed on the fimA Ⅱ and fimA Ⅳ genotypes). Please revise thoroughly.

Abstract

- Not acceptable in the present form. This reviewer indeed is somewhat astonished about the low quality of this revised draft. Please see, for example:

* ( P.gingivalis) must read (P. gingivialis) (must be italicized). Remember to define abbreviations upon first appearance in the text, then use the abbreviated forms.

* Please stick to PLOS ONE's Guidelines for Authors: "The Abstract — which must be no more than 300 words long and contain no references — should serve both as a general introduction to the topic and as a brief, non-technical summary of the main results and their implications." Ignoring these recommendations must lead to outright reject, in particular with a re-submitted draft. Please note that this part must be a stand-alone section, enabling future readers to switch to your full text. Provide as much information as possible here, but do not repeat again and again multiple aspects. For example, "P. gingivitis" has been mentioned 11 (!) times.

- Double check for grammar and style.

- What is "Martials"?

- Stick to the established sections and their respective names given with the Authors' Guidelines.

- Delete blank spaces and hard paragraphs marks considered dispensable.

- In contrast, "periodontaldisease" must read "periodontal disease". Same with "fimA Ⅳand", "fica ⅤPorphyromonas", "respectively.Conclusion", fimAⅡand". Please revise thoroughly, and remember that authors are responsible for procuring copy editing or language editing services for their manuscripts. Obtaining this service is the responsibility of the author, and should be done before initial submission (but NOT after the re-submission).

Intro

- Same aspects as given above would be evident with the full text sections. This revised and re-submitted draft would not seem ready to proceed. Major revisions would seem mandatory.

- Please remember that each and every statement needs a reference. See, for example, "Periodontitis is a chronic inflammation (...) characterized by the progressive loss of attachment and alveolar bone absorption." Reference missing.

- Again, authors are strongly encouraged to visit a recent paper published by PLOS ONE. There you will see that referencing does follow a format like this: "(...) interdental hygiene [6, 9], even with high-risk (...) [10, 11]. In fact, this approach (...) [12]." This also is clearly stated with the Guidelines for Authors, so please revise thoroughly throughout your text.

- See last recommendation: "Again, please elaborate your aims, and give your objectives." Obviously, the authors did not want to follow here.

- See last recommendation: "Deduce a sound and indisputable null hypothesis from your foregoing elaboration." Obviously, the authors did not want to follow here.

Meths

- From this revised draft it would seem clear that only two of the included studies were rated "high quality". This is considered poor.

- What about possible bias of the included studies?

- While some scientists occasionally have considered the systematic review (SR) as the top level of evidence for clinical science research in the past, please note that SR per se cannot be endorsed solely due to their format ("because they are a systematic review"), but must be critically evaluated for the problem reviewed in the respective article. Consequently, different qualities of RCTs, explicit statements of protocols, careful estimates of risk bias, large numbers of studies, and sophisticated statistical meta-analyses are not considered true substitutes for a thorough understanding of the literature, for critical analysis, and for scientifically sound and clinically relevant conclusions.

Results

- Providing several forest plots without guiding the reader would not seem acceptable.

Disc

- This section still has not been convincingly elaborated.

- Please do not simply repeat your results here. Instead, provide an intellectual discourse.

- Remember that section heading is NOT "Repetition". Provide a critical analysis of your findings. Provide interpretations. Discuss limitations.

- Please see above, and stick to the weaknesses of the included papers, give explanations and arguments.

- What about your null hypothesis?

Concl

- "This study showed that the fimA Ⅱand fimA Ⅳgenotypes of Porphyromonas gingivalis are

highly present in patients with periodontal disease." First, this already has been known before. Second (and again), do not simply repeat your results here. Instead, provide a reasonable extension of your outcome, and try to find some generalizing statements.

Refs

- Again, the authors obviously have failed to revise this section for a constant formatting, and this would seem surprising with a revised draft. Thorough revision still is mandatory, but, again, this would not seem the reviewer's task.

In total, this revised and re-submited paper draft is considered neither convincing nor satisfying. Please note that all authors are expected to have reviewed, discussed, and agreed to their individual contributions ahead of (re-)submission. Authors' „approval“ means that the (co-)author confirms the (co-)author has made a significant scientific contribution to the study and that he is thoroughly familiar with the primary data outlined in the manuscript, and that he has read and revised the complete manuscript, takes responsibility for the content and completeness of the final submitted manuscript.

Reviewer #2: Why didn't you adjust for factors such as smoking, alcohol , age and gender? Also tsome spaces between the words were ignored. Please make the correction.

Reviewer #3: Introduction:

1. Spacing is needed after the comma : Porphyromonas

gingivalis,Aggregatibacte actinomycetemcomitans,

2. Aggregatibacte actinomycetemcomitans: Spelling mistake: Correct would be Aggregatibacter actinomycetemcomitans

3. Term P.gingivalis has been used only in few places in the introduction but the term Porphyromonas

gingivalis has been used is the most part of the manuscript. Keeping it uniform would better help the readers.

Letter A in Acknowledgements and Letter R in References needs to bold to make it uniform.

7. PLOS authors have the option to publish the peer review history of their article (what does this mean?). If published, this will include your full peer review and any attached files.

Reviewer #1: No

Reviewer #2: No

Reviewer #3: No

---

## [Author Response · Author response to Decision Letter 1]

5 Aug 2020

Dear Editors and Reviewers:

Thank you for your letter and for the reviewers'comments concerning our manuscript entitled“The prevalence of fimA genotypes of Porphyromonas gingivalis in patients with chronic periodontitis: a meta-analysis” (ID: PONE-D-20-06291R1). Those comments are all valuable and very helpful for revising and improving our paper, as well as the important guiding significance to our researches. We have studied comments carefully and have made correction which we hope meet with approval. We have polished the language of the full text, see supporting information for the certificate. Revised portion are marked in red in the paper. The main corrections in the paper and the responds to the reviewer’s comments are as flowing:

Responds to the reviewer’s comments:

Response to comment:1. If the authors have adequately addressed your comments raised in a previous round of review and you feel that this manuscript is now acceptable for publication, you may indicate that here to bypass the “Comments to the Author” section, enter your conflict of interest statement in the “Confidential to Editor” section, and submit your "Accept" recommendation.

Reviewer #1: (No Response)

Response: The author has dealt with the comments raised in the last round of review. Of course, there may be insufficient answers. The author will make supplementary amendments according to the opinions of this round.

Response to comment:2. Is the manuscript technically sound, and do the data support the conclusions?

Reviewer #1: No

Response: This study describe a technically sound piece of scientific research with data that supports the conclusions. A total of 167 studies were identified through the search process. The articles processed were based on abstracts, titles and full texts, with a total of 17 papers were included. The total sample size was 1389 individuals with samples collected from different countries. The conclusions of this study are drawn from the data extracted from the literature.

Response to comment:3. Has the statistical analysis been performed appropriately and rigorously?

Reviewer #2: No

Response: The statistical methods used in this study have been explained in detail in the statistical analysis part of the article. We used a statistical software package (Stata, version 11.0/mp, statacorp) to calculate the data analysis for each combined incidence estimate.

Response to comment:5. Is the manuscript presented in an intelligible fashion and written in standard English?

Reviewer #1: No

Response: The article has been revised in standard English.

Reviewer #1:

Response to comment:- This revised draft has been improved to some extent, no doubt. However, there are still minor and major drawbacks, and this re-submitted version is not considered ready to proceed.

- Again, the authors are strongly encouraged to seek help from a native speaker experienced with scientific writing.

Response: We are very sorry for our negligence of the minor and major drawbacks in this draft ,according that we revised the draft thoroughly. In addition, we seek the help from the native speaker to revise our draft. 

Response to comment:- Title - "The prevalence of fimA genotype of Porphyromonas gingivalis in patients with chronic periodontitis :a meta-analysis". Please revise for correct punctuation. Remember that it is not considered the reviewers' task to co-author and/or re-edit your submitted draft.

- Go for italics with your species names.

Response: We are very sorry for our incorrect writing of the title. We have revised the title's punctuation and italics with species names.

Response to comment:- The term "prevalence of fimA genotype of Porphyromonas gingivitis" would seem misleading. Remember that you did not study other genotypes than "fimA". (Indeed, you have focussed on the fimA Ⅱ and fimA Ⅳ genotypes). Please revise thoroughly.

Response: Thank you for your kind reminder. We have performed the genotyping analyses for fimA I, II, III, IV and V. The results were shown in Fig. 2 through Fig. 7.

Response to comment: Abstract -- Not acceptable in the present form. This reviewer indeed is somewhat astonished about the low quality of this revised draft. Please see, for example:

* ( P.gingivalis) must read (P. gingivialis) (must be italicized). Remember to define abbreviations upon first appearance in the text, then use the abbreviated forms.

Response: We have made correction according to the Reviewer’s comments.

Response to comment:* Please stick to PLOS ONE's Guidelines for Authors: "The Abstract — which must be no more than 300 words long and contain no references — should serve both as a general introduction to the topic and as a brief, non-technical summary of the main results and their implications." Ignoring these recommendations must lead to outright reject, in particular with a re-submitted draft. Please note that this part must be a stand-alone section, enabling future readers to switch to your full text. Provide as much information as possible here, but do not repeat again and again multiple aspects. For example, "P. gingivitis" has been mentioned 11 (!) times.

Response: We have made correction according to the Reviewer’s comments.

Response to comment:- Double check for grammar and style.

Response: We have made correction according to the Reviewer’s comments.

Response to comment:- What is "Martials"?

Response: We have made correction according to the Reviewer’s comments.

Response to comment:- Stick to the established sections and their respective names given with the Authors' Guidelines.

Response: We have made correction according to the Reviewer’s comments.

Response to comment:- Delete blank spaces and hard paragraphs marks considered dispensable.

Response: We have made correction according to the Reviewer’s comments.

Response to comment:- In contrast, "periodontaldisease" must read "periodontal disease". Same with "fimA Ⅳand", "fica ⅤPorphyromonas", "respectively.Conclusion", fimAⅡand". Please revise thoroughly, and remember that authors are responsible for procuring copy editing or language editing services for their manuscripts. Obtaining this service is the responsibility of the author, and should be done before initial submission (but NOT after the re-submission).

Response: We are very sorry for our incorrect writing . We have made correction according to the Reviewer’s comments.

Response to comment: Intro - Same aspects as given above would be evident with the full text sections. This revised and re-submitted draft would not seem ready to proceed. Major revisions would seem mandatory.

- Please remember that each and every statement needs a reference. See, for example, "Periodontitis is a chronic inflammation (...) characterized by the progressive loss of attachment and alveolar bone absorption." Reference missing.

Response: We have made correction according to the Reviewer’s comments.

Response to comment:- Again, authors are strongly encouraged to visit a recent paper published by PLOS ONE. There you will see that referencing does follow a format like this: "(...) interdental hygiene [6, 9], even with high-risk (...) [10, 11]. In fact, this approach (...) [12]." This also is clearly stated with the Guidelines for Authors, so please revise thoroughly throughout your text.

Response: We have made correction according to the Reviewer’s comments.

Response to comment:- See last recommendation: "Again, please elaborate your aims, and give your objectives." Obviously, the authors did not want to follow here.

Response: We have made correction according to the Reviewer’s comments.

Response to comment:- See last recommendation: "Deduce a sound and indisputable null hypothesis from your foregoing elaboration." Obviously, the authors did not want to follow here.

Response: We have made correction according to the Reviewer’s comments.

Response to comment: Meths - From this revised draft it would seem clear that only two of the included studies were rated "high quality". This is considered poor.

Response: Thank you for your comments. Yes. There are less high-quality original articles used for the meta-analysis. So we performed the further analysis to compare the high-quality and low-quality articles.

Response to comment:- What about possible bias of the included studies?

Response: There may be publication bias, that is, there may be negative results not published.

Response to comment:- While some scientists occasionally have considered the systematic review (SR) as the top level of evidence for clinical science research in the past, please note that SR per se cannot be endorsed solely due to their format ("because they are a systematic review"), but must be critically evaluated for the problem reviewed in the respective article. Consequently, different qualities of RCTs, explicit statements of protocols, careful estimates of risk bias, large numbers of studies, and sophisticated statistical meta-analyses are not considered true substitutes for a thorough understanding of the literature, for critical analysis, and for scientifically sound and clinically relevant conclusions.

Response: Yes. I very agree with you. The meta-analysis is only the analysis of more data by collecting the related studies when there are less clear results. The aim of meta-analysis is for higher power of statistical analysis. But the results were affected by the quality of the original article. In fact, it should be best for RCT with large participants, just as the RECOVERY Study done by Oxford University for the study of COVID-19.

Response to comment: Results- Providing several forest plots without guiding the reader would not seem acceptable.

Response: We have made correction according to the Reviewer’s comments.

Response to comment: Disc- This section still has not been convincingly elaborated.

- Please do not simply repeat your results here. Instead, provide an intellectual discourse.

- Remember that section heading is NOT "Repetition". Provide a critical analysis of your findings. Provide interpretations. Discuss limitations.

- Please see above, and stick to the weaknesses of the included papers, give explanations and arguments.

- What about your null hypothesis?

Response: Considering the Reviewer’s suggestion, We have re-written this part according to the Reviewer’s suggestion.

Response to comment: Refs- Again, the authors obviously have failed to revise this section for a constant formatting, and this would seem surprising with a revised draft. Thorough revision still is mandatory, but, again, this would not seem the reviewer's task.

Response: We have made correction according to the Reviewer’s comments.

Reviewer #2:

Response to comment: Why didn't you adjust for factors such as smoking, alcohol , age and gender? Also some spaces between the words were ignored. Please make the correction.

Response: Thank you for your comments. Yes. We should adjust the smoking, alcohol , age and gender, but the data were inadequate. We have corrected the spaces between the words. 

Reviewer #3:

Response to comment: Introduction:

1. Spacing is needed after the comma : Porphyromonas gingivalis,Aggregatibacte actinomycetemcomitans,

Response: We have made correction according to the Reviewer’s comments.

Response to comment: 2. Aggregatibacte actinomycetemcomitans: Spelling mistake: Correct would be Aggregatibacter actinomycetemcomitans

Response: We have made correction according to the Reviewer’s comments.

Response to comment:3. Term P.gingivalis has been used only in few places in the introduction but the term Porphyromonas gingivalis has been used is the most part of the manuscript. Keeping it uniform would better help the readers.

Response: We have made correction according to the Reviewer’s comments.

Response to comment: Letter A in Acknowledgements and Letter R in References needs to bold to make it uniform.

Response: We have made correction according to the Reviewer’s comments.

Response to comment:7. PLOS authors have the option to publish the peer review history of their article (what does this mean?). If published, this will include your full peer review and any attached files.

Do you want your identity to be public for this peer review? For information about this choice, including consent withdrawal, please see our Privacy Policy.

Reviewer #1: No

Reviewer #2: No

Reviewer #3: No

Response: We have made correction according to the Reviewer’s comments.

Thank you！

---

## [Decision Letter · Decision Letter 2]

17 Aug 2020

PONE-D-20-06291R2

The prevalence of fimA genotypes of Porphyromonas gingivalis in patients with chronic periodontitis: a meta-analysis

PLOS ONE

Dear Dr. Wang,

Thank you for submitting your manuscript to PLOS ONE. After careful consideration, we feel that it has merit but does not fully meet PLOS ONE’s publication criteria as it currently stands. Therefore, we invite you to submit a revised version of the manuscript that addresses the points raised during the review process.

Having intensively reviewed your draft, your revised and re-submitted draft still would not seem satisfying. I have double checked your submitted draft, and, in particular, you should follow the R #1 comments, to finalize your paper convincingly, and to meet both PLOS ONE's quality standards and our readership's expectations. Please note that a further non-convincing revision (not considered acceptable with regard to language, reviewers' constructive criticism, content, generalizable outcome, and/or Authors' Guidelines) will lead to outright reject.

We look forward to receiving your revised manuscript.

Kind regards,

Andrej M Kielbassa

Academic Editor

PLOS ONE

Reviewers' comments:

Reviewer's Responses to Questions

**Comments to the Author**

1. If the authors have adequately addressed your comments raised in a previous round of review and you feel that this manuscript is now acceptable for publication, you may indicate that here to bypass the “Comments to the Author” section, enter your conflict of interest statement in the “Confidential to Editor” section, and submit your "Accept" recommendation.

Reviewer #1: (No Response)

Reviewer #2: All comments have been addressed

2. Is the manuscript technically sound, and do the data support the conclusions?

Reviewer #1: No

Reviewer #2: Yes

3. Has the statistical analysis been performed appropriately and rigorously? 

Reviewer #1: Yes

Reviewer #2: Yes

4. Have the authors made all data underlying the findings in their manuscript fully available?

Reviewer #1: Yes

Reviewer #2: Yes

5. Is the manuscript presented in an intelligible fashion and written in standard English?

Reviewer #1: No

Reviewer #2: Yes

6. Review Comments to the Author

Reviewer #1: Abstract

- Please change sequence: "A total of 17 studies were included in this report." should follow after "The reference lists of relevant published articles were searched manually."

Intro

- PMID number never will appear with your text, and this reviewer clearly is wondering about the authors' intention. Again, it should be emphasized that having a closer look into recently published PLOS ONE papers would be advantageous. Please delete "( PMID:23991022 )" and all other PMIDs. These should appear with your Ref section.

- Same with your Ref numbers. This must be separated by using the spacebar. See, for example: "(...) can lead to the loss of affected teeth [1]." Please note that this should not be necessary with a third (re-)submission, and, to be honest, the authors should re-think their attitude towards their willingness to adequately revise their draft. Remember that with your re-submission, you affirm that ALL FOUR CO-AUTHORS have read and approved this revised draft.

- What do you refer to when stating "In this study, (...)"? Do you mean the present study, or do you refer to [14]? Again, please revise thoroughly, to facilitate reading.

- Again, please elaborate both aims and objectives more clearly. Remember that aims are statements of intent. They are usually written in broad terms. They set out what you hope to achieve at the end of the project. Objectives, on the other hand, should be specific statements that define measurable outcomes, e.g. what steps will be taken to achieve the desired outcome (see https://learn.solent.ac.uk/mod/book/view.php?id=116233&chapterid=15294).

- Finally, a clear and indisputable null hypothesis is missing. Please remember that H0 must be deducible from the foregoing thoughts.

Please note that this section still would not seem convincing.

Meths

- "All articles with the keywords were (...)." How many articles are you talking about?

- "Studies were excluded (...)." How many articles are you talking about? Please add numbers.

- "(...) assessment tool.[28] We assessed (...)." Why do you superscript the Ref number here?

- "The total score for each study was between 0 and 8." This would be self-explaining. What about providing a complete analysis, thus revealing the exact quality of the various studies?

Results

- "(...) and therefore the results of most related studies were included in the analysis." Most, but not all, right? Again, please add complete and exact information, and do not stick to vague statements.

Disc

- Please stick to H0 when staring this section.

- "(...) in mouse models[31](PMID: 17081195)[36] and (...)." Please see comments given above. Make use of your spacebar to separate Refs, delete PMID, and do not superscript Refs.

- "Presently, there is no other meta-analysis similar to this research in terms of ability to compare results." One sentence does not constitute one paragraph.

- "First, publication bias is an inevitable problem in a meta-analysis process." This should be discussed more thoroughly. Same with the other aspects.

Concl

- "In conclusion, P. gingivalis is one of main bacteria in chronic periodontitis." This aspect is not considered a conclusion deducible FROM YOUR STUDY.

- Same with "These results indicate that the pathogenicity of other fimA genotypes of P. gingivalis needs to be further studied." This might be widely spread, but, to be honest, would be a simple stereotype only.

Refs

- Please note that your Refs still have not been uniformly formatted. Again, please check Guidelines, and consult some recent papers published in PLOS ONE.

In total, this re-revised and re-re-submitted draft is, unfortunately, not ready to proceed.

Please note that SR per se cannot be endorsed solely due to their format ("because they are a systematic review"), but must be critically evaluated for the problem reviewed in the respective article. Consequently, different qualities of RCTs (as given with your study, please see below), explicit statements of protocols, careful estimates of risk bias, large numbers of studies, and sophisticated statistical meta-analyses are not considered true substitutes for a thorough understanding of the literature, for critical analysis, and for scientifically sound and clinically relevant conclusions considered generalizable.The authors might additionally wish to go to https://pubmed.ncbi.nlm.nih.gov/31088221/, and discuss these problems.

Reviewer #2: (No Response)

7. PLOS authors have the option to publish the peer review history of their article (what does this mean?). If published, this will include your full peer review and any attached files.

Reviewer #1: No

Reviewer #2: No

---

## [Author Response · Author response to Decision Letter 2]

11 Sep 2020

Dear Editors and reviewers:

Thank you for your letter and for the reviewers'comments concerning our manuscript entitled "The prevalence of fimA genotypes of Porphyromonas gingivalis in patients with chronic periodontitis: a meta-analysis" (ID: PONE-D-20-06291R1). Those comments are all valuable and very helpful for revising and improving our paper, as well as the important guiding significance to our researches. We have studied comments carefully and have made correction which we hope meet with approval. Revised portion are marked in red in the paper. The main corrections in the paper and the responds to the reviewer’s comments are as flowing:

 Responds to the reviewer’s comments:

 1. If the authors have adequately addressed your comments raised in a previous round of review and you feel that this manuscript is now acceptable for publication, you may indicate that here to bypass the “Comments to the Author” section, enter your conflict of interest statement in the “Confidential to Editor” section, and submit your "Accept" recommendation.

 Reviewer #1: (No Response)

 Response: The author has dealt with the comments raised in the last round of review. Of course, there may be insufficient answers. The author will make supplementary amendments according to the opinions of this round.

2. Is the manuscript technically sound, and do the data support the conclusions?

Reviewer #1: No

Response: This study describe a technically sound piece of scientific research with data that supports the conclusions. We evaluated the quality of the 17 included literatures and all met the requirements. The conclusions of this study are drawn from the data extracted from the literature.

5. Is the manuscript presented in an intelligible fashion and written in standard English?

Reviewer #1: No

Response: Thank you for your comments. We have revised our article in standard English.

- Please change sequence: "A total of 17 studies were included in this report." should follow after "The reference lists of relevant published articles were searched manually."

Response: Thank you for your comments. We have made correction according to the Reviewer’s comments.

- PMID number never will appear with your text, and this reviewer clearly is wondering about the authors' intention. Again, it should be emphasized that having a closer look into recently published PLOS ONE papers would be advantageous. Please delete "( PMID:23991022 )" and all other PMIDs. These should appear with your Ref section.

Response: Thank you for your comments.We have made correction according to the Reviewer’s comments.

- Same with your Ref numbers. This must be separated by using the spacebar. See, for example: "(...) can lead to the loss of affected teeth [1]." Please note that this should not be necessary with a third (re-)submission, and, to be honest, the authors should re-think their attitude towards their willingness to adequately revise their draft. Remember that with your re-submission, you affirm that ALL FOUR CO-AUTHORS have read and approved this revised draft.

Response: Thank you for your comments. We have made correction according to the Reviewer’s comments.

- What do you refer to when stating "In this study, (...)"? Do you mean the present study, or do you refer to [14]? Again, please revise thoroughly, to facilitate reading.

Response: Thank you for your comments. We have made correction according to the Reviewer’s comments.

- Again, please elaborate both aims and objectives more clearly. Remember that aims are statements of intent. They are usually written in broad terms. They set out what you hope to achieve at the end of the project. Objectives, on the other hand, should be specific statements that define measurable outcomes, e.g. what steps will be taken to achieve the desired outcome.

Response: Thank you for your comments. We have made correction according to the Reviewer’s comments.

- Finally, a clear and indisputable null hypothesis is missing. Please remember that H0 must be deducible from the foregoing thoughts.

Please note that this section still would not seem convincing.

Response: Thank you for your comments. We have made correction according to the Reviewer’s comments.

- "All articles with the keywords were (...)." How many articles are you talking about?

Response: Thank you for your comments. We have made correction according to the Reviewer’s comments.

- "Studies were excluded (...)." How many articles are you talking about? Please add numbers.

Response: Thank you for your comments. We have made correction according to the Reviewer’s comments.

- "(...) assessment tool.[28] We assessed (...)." Why do you superscript the Ref number here?

- "The total score for each study was between 0 and 8." This would be self-explaining. What about providing a complete analysis, thus revealing the exact quality of the various studies?

Response: Thank you for your comments. The purpose of marking the reference number here is to better explain to readers the basis of literature scoring through references. Of course, the author also revised table 1 to list the detailed scoring details of each literature, thus revealing the exact quality of various studies. Refer to and use effective quality evaluation standards to evaluate the quality of the included literature (scoring basis): (1 point for meeting one item)

(1) Clear definition of target population

(2) Representativeness of sampling

(3) The matching of samples and the whole population

(4) Sufficient response rate

(5) Standardized data collection methods

(6) Reliable investigation method

(7) Effective investigation measures

(8) Appropriate statistical methods

- "(...) and therefore the results of most related studies were included in the analysis." Most, but not all, right? Again, please add complete and exact information, and do not stick to vague statements.

Response: Thank you for your comments. We have made correction according to the Reviewer’s comments.

- Please stick to H0 when staring this section.

Response: Thank you for your comments. We have made correction according to the Reviewer’s comments.

- "(...) in mouse models[31](PMID: 17081195)[36] and (...)." Please see comments given above. Make use of your spacebar to separate Refs, delete PMID, and do not superscript Refs.

Response: Thank you for your comments. We have made correction according to the Reviewer’s comments.

- "Presently, there is no other meta-analysis similar to this research in terms of ability to compare results." One sentence does not constitute one paragraph.

Response: Thank you for your comments. We have made correction according to the Reviewer’s comments.

- "First, publication bias is an inevitable problem in a meta-analysis process." This should be discussed more thoroughly. Same with the other aspects.

Response: Thank you for your comments. We have made correction according to the Reviewer’s comments.

- "In conclusion, P. gingivalis is one of main bacteria in chronic periodontitis." This aspect is not considered a conclusion deducible FROM YOUR STUDY.

Response: Thank you for your comments. We have made correction according to the Reviewer’s comments.

- Same with "These results indicate that the pathogenicity of other fimA genotypes of P. gingivalis needs to be further studied." This might be widely spread, but, to be honest, would be a simple stereotype only.

Response: Thank you for your comments. We have made correction according to the Reviewer’s comments.

- Please note that your Refs still have not been uniformly formatted. Again, please check Guidelines, and consult some recent papers published in PLOS ONE.

Response: Thank you for your comments. We have made correction according to the Reviewer’s comments.

 Thank you!

---

## [Decision Letter · Decision Letter 3]

23 Sep 2020

The prevalence of fimA genotypes of Porphyromonas gingivalis in patients with chronic periodontitis: a meta-analysis

PONE-D-20-06291R3

Dear Dr. Wang,

We’re pleased to inform you that your manuscript has been judged scientifically suitable for publication and will be formally accepted for publication once it meets all outstanding technical requirements.

Kind regards, and stay healthy, please

Andrej M Kielbassa, Prof. Dr. med. dent. Dr. h. c.

Academic Editor

PLOS ONE

Additional Editor Comments (optional):

This paper is ready to proceed.

Reviewers' comments:

Reviewer's Responses to Questions

**Comments to the Author**

1. If the authors have adequately addressed your comments raised in a previous round of review and you feel that this manuscript is now acceptable for publication, you may indicate that here to bypass the “Comments to the Author” section, enter your conflict of interest statement in the “Confidential to Editor” section, and submit your "Accept" recommendation.

Reviewer #1: All comments have been addressed

Reviewer #2: All comments have been addressed

2. Is the manuscript technically sound, and do the data support the conclusions?

Reviewer #1: Yes

Reviewer #2: Yes

3. Has the statistical analysis been performed appropriately and rigorously? 

Reviewer #1: Yes

Reviewer #2: Yes

4. Have the authors made all data underlying the findings in their manuscript fully available?

Reviewer #1: Yes

Reviewer #2: Yes

5. Is the manuscript presented in an intelligible fashion and written in standard English?

Reviewer #1: Yes

Reviewer #2: Yes

6. Review Comments to the Author

Reviewer #1: Still there are some minor shortcomings with the written text. These aspects will be perfectible with the proofs. This paper is acceptable now.

Reviewer #2: the authors addressed all the questions.The paper can be published

7. PLOS authors have the option to publish the peer review history of their article (what does this mean?). If published, this will include your full peer review and any attached files.

Reviewer #1: No

Reviewer #2: No

---

## [Editor Report · Acceptance letter]

25 Sep 2020

PONE-D-20-06291R3 

The prevalence of *fimA* genotypes of Porphyromonas gingivalis in patients with chronic periodontitis: a meta-analysis 

Dear Dr. Wang:

I'm pleased to inform you that your manuscript has been deemed suitable for publication in PLOS ONE. Congratulations! Your manuscript is now with our production department. 

Kind regards, 

on behalf of

Prof. Dr. med. dent. Dr. h. c. Andrej M Kielbassa 

Academic Editor

PLOS ONE